# A Fast, Robust Elliptical Slice Sampling Method for Truncated Multivariate Normal Distributions

**Kaiwen Wu**
University of Pennsylvania
kaiwenwu@seas.upenn.edu

**Jacob R. Gardner**
University of Pennsylvania
jacobrg@seas.upenn.edu

## Abstract

Elliptical slice sampling, when adapted to linearly truncated multivariate normal distributions, is a rejection-free Markov chain Monte Carlo method. At its core, it requires analytically constructing an ellipse-polytope intersection. The main novelty of this paper is an algorithm that computes this intersection in $\mathcal{O}(m \log m)$ time, where $m$ is the number of linear inequality constraints representing the polytope. We show that an implementation based on this algorithm enhances numerical stability, speeds up running time, and is easy to parallelize for launching multiple Markov chains.

## 1 Introduction

Let $\mathbf{x} \sim \mathcal{N}(\mathbf{0}, \mathbf{I})$ be a $d$-dimensional standard normal random variable. This paper is concerned with sampling from the truncated multivariate normal distribution

$$p(\mathbf{x}) = \begin{cases} \frac{1}{Z}\phi(\mathbf{x}) & \mathbf{x} \in \mathcal{D}, \\ 0 & \mathbf{x} \notin \mathcal{D}, \end{cases}$$

where $\phi(\mathbf{x}) \propto \exp\left(-\frac{1}{2}\mathbf{x}^\top \mathbf{x}\right)$ is the standard normal density, $Z = \int_{\mathbf{x} \in \mathcal{D}} \phi(\mathbf{x})\, \mathrm{d}\mathbf{x}$ is a normalization constant, and the domain $\mathcal{D} = \{\mathbf{x} \in \mathbb{R}^d : \mathbf{A}\mathbf{x} \leq \mathbf{b}\}$ is a polytope defined by $m$ linear inequalities with $\mathbf{A} \in \mathbb{R}^{m \times d}$ and $\mathbf{b} \in \mathbb{R}^m$. We assume the polytope domain has a non-empty interior but is not necessarily bounded. The standard normal assumption is without loss of generality, since non-standard normal distributions can be handled by a change of variable, as shown in §A.

Truncated normal sampling has numerous applications in machine learning and statistics, with recent ones in skew Gaussian processes (e.g., Benavoli et al., 2021) and preferential Bayesian optimization (Benavoli et al., 2021; Takeno et al., 2023). In addition, truncated normal sampling is a key building block of sophisticated numerical methods estimating integrals related to truncated normal distributions (Gessner et al., 2020).

Linear elliptical slice sampling (e.g., Murray et al., 2010; Fagan et al., 2016; Gessner et al., 2020) is a rejection-free tuning-free Markov chain Monte Carlo method for truncated normal distributions. Each iteration analytically constructs the intersection of an ellipse and the polytope domain, from which the next iterate is sampled. In principle, this method is particularly suitable for high dimensional truncated normal distributions thanks to its rejection free property.

However, the devil is in the details. Despite its conceptual simplicity, constructing the ellipse-polytope intersection is easier said than done. We will show that all existing implementations share a worst-case time complexity of $\mathcal{O}(m^2)$, which scales poorly as the number of constraints increases. Moreover, existing implementations have complex control flows, which makes it hard, if not impossible, to parallelize on GPUs. Indeed, to the best of our knowledge, there is no batch implementation of elliptical slice sampling to this date, which is likely due to the programming complexity.

Workshop on Bayesian Decision-making and Uncertainty, 38th Conference on Neural Information Processing Systems (NeurIPS 2024).

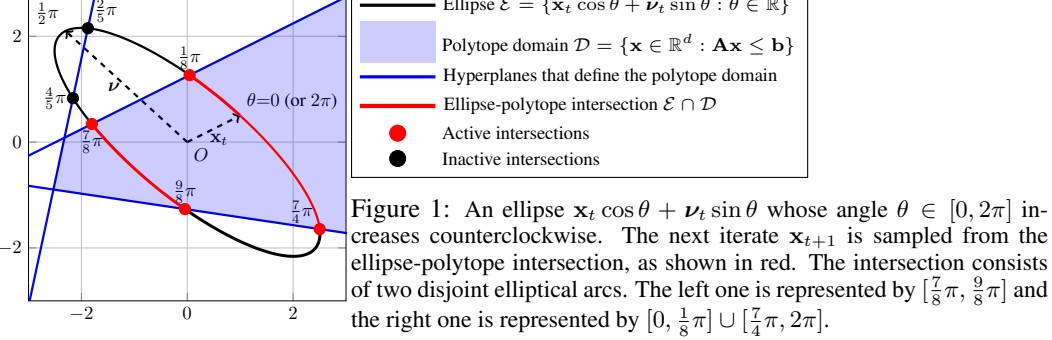

Figure 1: An ellipse $\mathbf{x}_t \cos\theta + \boldsymbol{\nu}_t \sin\theta$ whose angle $\theta \in [0, 2\pi]$ increases counterclockwise. The next iterate $\mathbf{x}_{t+1}$ is sampled from the ellipse-polytope intersection, as shown in red. The intersection consists of two disjoint elliptical arcs. The left one is represented by $[\frac{7}{8}\pi, \frac{9}{8}\pi]$ and the right one is represented by $[0, \frac{1}{8}\pi] \cup [\frac{7}{4}\pi, 2\pi]$.

**Contributions.** We develop a new algorithm computing the ellipse-polytope intersection that has a better time complexity and is easier to implement. The algorithm runs in $\mathcal{O}(m \log m)$ time and is faster than the existing implementations. Moreover, this algorithm has a simple control flow and is particularly amenable to GPU parallelism. As a result, we are able to parallelize thousands of independent Markov chains easily. Experiments show that our implementation accelerates truncated normal sampling massively in high dimensions.

## 2 Elliptical Slice Sampling for Truncated Normal Sampling

In the $t$-th iteration of linear elliptical slice sampling (e.g., Fagan et al., 2016; Gessner et al., 2020), we sample a multivariate normal random variable $\boldsymbol{\nu}_t \sim \mathcal{N}(\mathbf{0}, \mathbf{I})$ and form an ellipse

$$\mathcal{E} = \{\mathbf{x}_t \cos\theta + \boldsymbol{\nu}_t \sin\theta : \theta \in \mathbb{R}\}. \tag{1}$$

The next iterate $\mathbf{x}_{t+1}$ is sampled from the ellipse-polytope intersection $\mathcal{E} \cap \mathcal{D}$, i.e., the parts of the ellipse that lie inside the polytope domain. This intersection can be constructed analytically by exploiting the polytope structure, and thus no rejection sampling is needed. See Figure 1 for an illustration and Algorithm 1 for the pseudocode. The arising questions are, of course, how to "analytically construct" the ellipse-polytope intersection $\mathcal{E} \cap \mathcal{D}$ and how to do it efficiently.

Note that the polytope domain itself is the intersection of $m$ halfspaces:

$$\mathcal{D} = \bigcap_{i=1}^{m} \mathcal{H}_i = \bigcap_{i=1}^{m} \{\mathbf{x} \in \mathbb{R}^d : \mathbf{a}_i^\top \mathbf{x} \le b_i\},$$

where $\mathbf{a}_i$ is the $i$-th row of $\mathbf{A}$. Thus, the ellipse-polytope intersection $\mathcal{E} \cap \mathcal{D}$ reduces to constructing each ellipse-halfspace intersection $\mathcal{E} \cap \mathcal{H}_i$, which does admit an analytical construction.

The intersection of the ellipse and the $i$-th halfspace $\mathcal{H}_i$ is an elliptical arc. The end points of the elliptical arc are identified by the intersection angles, i.e., the roots of the trigonometry equation

$$\mathbf{a}_i^\top \mathbf{x}_t \cos\theta + \mathbf{a}_i^\top \boldsymbol{\nu}_t \sin\theta = b_i, \tag{2}$$

which typically indeed has two distinct roots $\alpha_i$ and $\beta_i$ in closed-forms (see §B). Without loss of generality, we assume all intersection angles $\alpha_i$ and $\beta_i$ are converted into $[0, 2\pi]$ by, if necessary,

---

**Algorithm 1:** Elliptical Slice Sampling for Truncated Multivariate Normal Distributions

1   Initialize $\mathbf{x}_0 \in \mathcal{D}$
2   **for** $t = 1, 2, \cdots$ **do**
3      sample $\boldsymbol{\nu}_t \sim \mathcal{N}(\mathbf{0}, \mathbf{I})$ and form an ellipse $\mathcal{E} = \{\mathbf{x}_t \cos\theta + \boldsymbol{\nu}_t \sin\theta : \theta \in \mathbb{R}\}$
4      compute the active intervals $I_{\text{act}}$ corresponding to the ellipse-polytope intersection
5      sample uniformly $\theta \sim I_{\text{act}}$
6      $\mathbf{x}_{t+1} = \mathbf{x}_t \cos\theta + \boldsymbol{\nu}_t \sin\theta$
7   **end**

---

**Algorithm 2:** Constructing the Active Intervals Analytically

**Input:** $I_i = [0, \alpha_i] \cup [\beta_i, 2\pi]$ with $\alpha_i < \beta_i$ for $i = 1, 2, \cdots, m$
**Output:** $I_{\mathrm{act}} = \cap_{i=1}^m I_i$
1  sort $\{\alpha_i\}_{i=1}^m$ in ascending order: $0 \le \alpha_{i_1} \le \alpha_{i_2} \le \cdots \le \alpha_{i_m} \le 2\pi$
2  compute $\gamma_k = \max\{\beta_{i_1}, \beta_{i_2}, \cdots, \beta_{i_k}\}$ for $k = 1, 2, \cdots m$
3  **return** $[0, \alpha_{i_1}] \cup \left(\bigcup_{k=2}^m [\gamma_{k-1}, \alpha_{i_k}]\right) \cup [\gamma_m, 2\pi]$ // `define` $[\gamma_{k-1}, \alpha_{i_k}] = \emptyset$ `if` $\gamma_{k-1} > \alpha_{i_k}$

adding or subtracting a multiple of $2\pi$. In addition, we assume $\alpha_i$ is strictly smaller than $\beta_i$. A simple observation is that the ellipse-halfspace intersection $\mathcal{E} \cap \mathcal{H}_i$ is precisely represented by the union of two disjoint intervals:

$$I_i = [0, \alpha_i] \cup [\beta_i, 2\pi].$$

Note that the interval $I_i$ has two disjoint segments due to periodicity: The point $\mathbf{x}_t$ is represented by two distinct angles $0$ and $2\pi$. Intersecting all $I_i$'s gives the interval representation of the ellipse-polytope intersection:

$$I_{\mathrm{act}} = \bigcap_{i=1}^m I_i,$$

which we call the active intervals. Note that the plural form is used because $I_{\mathrm{act}}$ may consist of several disjoint intervals, each of which is an active interval. There is an one-to-one correspondence between angles in the active intervals (except the repetition of $\theta = 0$ and $\theta = 2\pi$) and points in the ellipse-polytope intersection.

Below we discuss existing methods constructing the active intervals.

**Brute Force.** The most straightforward algorithm follows the definition of the active intervals:

$$I_{\mathrm{act}}^{(0)} = [0, 2\pi], \quad I_{\mathrm{act}}^{(i)} = I_{\mathrm{act}}^{(i-1)} \cap \left([0, \alpha_i] \cup [\beta_i, 2\pi]\right).$$

In the end, $I_{\mathrm{act}}^{(m)}$ equals to the desired active intervals $I_{\mathrm{act}}$. However, this brute force intersection is tedious to implement and hard to parallelize. Moreover, its worst-case time complexity is as slow as $\mathcal{O}(m^2)$. See §D for why it is not $\mathcal{O}(m)$, contrary to how it may appear.

**Angle Perturbation.** In Figure 1, $\theta = \frac{2}{5}\pi$ and $\theta = \frac{4}{5}\pi$ are not active, since they do not contribute to the active intervals. The hard part of computing the active intervals precisely lies in identifying those active intersection angles. Gessner et al. (2020) identify the active angles by angle perturbations: An intersection angle $\theta$ is active if and only if exactly one of the perturbations $\theta_i - \epsilon$ and $\theta_i + \epsilon$ corresponds a point within the domain $\mathcal{D}$. The time complexity is $\mathcal{O}(m^2)$.

## 3  A Simple Method for the Ellipse-Polytope Intersection

Sorting $\alpha_i$'s in ascending order yields

$$0 \le \alpha_{i_1} \le \alpha_{i_2} \le \cdots \le \alpha_{i_k} \le \cdots \le \alpha_{i_m} \le 2\pi. \tag{3}$$
$$\begin{array}{cccc} \wedge & \wedge & \wedge & \wedge \\ \beta_{i_1} & \beta_{i_2} & \beta_{i_k} & \beta_{i_m} \end{array}$$

Note that $\beta_{i_k}$ is not necessarily monotonic in $k$. Our algorithm is based on the observation below.

**Proposition 1.** *Let $\alpha_i < \beta_i$ for all $i \in [m]$ and let $\{\alpha_{i_k}\}_{k=1}^m$ be sorted in ascending order as in* (3). *Then, the active intervals $I_{\mathrm{act}} = \cap_{i=1}^m \left([0, \alpha_i] \cup [\beta_i, 2\pi]\right)$ have an equivalent representation*

$$I_{\mathrm{act}} = [0, \alpha_{i_1}] \cup \left(\bigcup_{k=2}^m [\gamma_{k-1}, \alpha_{i_k}]\right) \cup [\gamma_m, 2\pi],$$

*where $\gamma_k = \max\{\beta_{i_1}, \beta_{i_2}, \cdots, \beta_{i_k}\}$ is the cumulative max until $\beta_{i_k}$. We interpret the interval $[\gamma_{k-1}, \alpha_{i_k}]$ as an empty set if $\gamma_{k-1} > \alpha_{i_k}$.*

Proposition 1 gives a closed-form expression for the active intervals $I_{\mathrm{act}}$, which yields Algorithm 2. Despite a somewhat lengthy proof, the idea and the final expression are both very simple. The time complexity is $\mathcal{O}(m \log m)$, faster than the brute force algorithm and likelihood testing. In addition, it is simple to program, amenable to GPU parallelism, and easy to batch, since the sorting and cumulative max operations are well supported in every popular machine learning package nowadays.

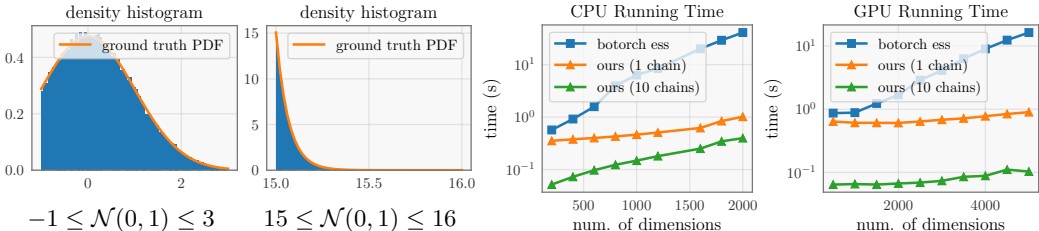

Figure 2: **1st and 2nd subfigures:** Draw $10^5$ samples from two univariate truncated normal distributions with parallel Markov chains. **3rd and 4th subfigures:** Compare running time of drawing 1000 samples from high dimensional truncated normal distributions.

## 4   Experiments

All simulations are run on a single machine with a RTX 3090 GPU using single precision floating points. We use the linear elliptical slice sampling implementation in BoTorch v0.11.1 as the baseline. Our code will be released publicly.

### 4.1   Sanity Check: One Dimensional Truncated Normal Sampling

We run elliptical slice sampling with Algorithm 2 on two univariate truncated normal distributions: $\mathcal{N}(0, 1)$ truncated by $-1 \leq x \leq 3$ and $15 \leq x \leq 16$, respectively. We draw $10^5$ samples from each distribution by running 2000 independent Markov chains in parallel with 500 iterations of burn-in and a thinning of 10. Hence, the total number of steps is $2000 \times 500 + 10 \times 10^5 = 2 \times 10^6$. The mean and variance estimates are accurate at least to the second digit after the decimal point (see §E).

### 4.2   Accelerate High Dimensional Truncated Normal Sampling

We demonstrate Algorithm 2 accelerates high dimensional truncated normal sampling, especially when the number of inequality constraints $m$ is large. We generate a set of random instances with varying dimensions as follows. First, we generate a $d \times d$ random matrix $\mathbf{A}$ whose entries are *i.i.d.* samples from a univariate standard normal distribution. Second, we generate a random vector $\mathbf{x}_0$ drawn from a $d$-dimensional standard normal distribution, which will used as the initialization the Markov chain. Third, we set $\mathbf{b} = \mathbf{A}\mathbf{x}_0 + \mathbf{u}$, where $\mathbf{u}$ is a random vector drawn uniformly from the hypercube $[0, 1]^d$. By construction, the initialization $\mathbf{x}_0$ lies in the interior of the domain. Note that the number of constraints $m = d$ increases as the number of dimensions increases.

In Figure 2, we draw 1000 samples from the general instances of truncated normal distributions and compare the running time against BoTorch's implementation. BoTorch's implementation runs a single Markov chain for 1000 steps. Our implementation runs either a single Markov chain for 1000 steps or 10 chains in parallel for 100 steps. Both of them use no burn-in and no thinning. No rejection occurs when running our implementation on these high dimensional distributions. With a single Markov chain, our implementation is over 10x faster than BoTorch's implementation in high dimensions, e.g., $d \geq 1000$ on CPU and $d \geq 4000$ on GPU. This speed-up solely comes from the improved per iteration complexity $\mathcal{O}(m \log m)$. Furthermore, running 10 Markov chains on GPU in parallel yields an additional 10x speed-up in high dimensions.

## 5   Conclusion

We have presented a $\mathcal{O}(m \log m)$ algorithm computing the active intervals in linear elliptical slice sampling for linearly truncated normal distributions. We hope our algorithm and implementation unlock the full potential of elliptical slice sampling for linearly truncated normal distributions, and enable new applications that are previously bottlenecked by the speed of sampling.

We end this paper by mentioning two extensions. First, it is interesting to adapt elliptical slice sampling to handle linear equality constraints, in which case the Markov chain has to run in the null space of the linear equality constraints. Second, it is interesting to support differentiable samples by adapting the idea of Zoltowski et al. (2021).

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

## A  Non-Standard Normal Distributions

The standard normal assumption is without loss of generality, since non-standard normal distributions reduce to the standard one by a change of variable. Let $\mathbf{x} \sim \mathcal{N}(\boldsymbol{\mu}, \boldsymbol{\Sigma})$ and let $\mathbf{L}\mathbf{L}^\top = \boldsymbol{\Sigma}$ be the Cholesky decomposition. Let $\mathbf{u} \sim \mathcal{N}(\mathbf{0}, \mathbf{I})$ be a standard normal variable. Truncating $\mathbf{x}$ by $\mathcal{D} = \{\mathbf{x} \in \mathbb{R}^d : \mathbf{A}\mathbf{x} \le \mathbf{b}\}$ is the same as truncating $\mathbf{u}$ by a transformed domain $\mathcal{D}' = \{\mathbf{u} \in \mathbb{R}^d : \mathbf{A}\mathbf{L}\mathbf{u} \le \mathbf{b} - \mathbf{A}\boldsymbol{\mu}\}$. Thus, we can sample from the truncated standard normal, truncated by $\mathcal{D}'$, and then apply a linear transformation $\mathbf{u} \mapsto \mathbf{L}\mathbf{u} + \boldsymbol{\mu}$.

## B  Roots of the Trigonometry Equation

This section solves the trigonometry equation

$$\mathbf{a}^\top \mathbf{x} \cos\theta + \mathbf{a}^\top \boldsymbol{\nu} \sin\theta = b.$$

Define $p = \mathbf{a}^\top \mathbf{x}$, $q = \mathbf{a}^\top \boldsymbol{\nu}$, and $r = \sqrt{p^2 + q^2}$. WLOG, we assume $r \ne 0$, otherwise the corresponding inequality constraint is either invalid ($b < 0$) or a tautology ($b \ge 0$). Note that the ratio $b/r \ge -1$, otherwise we have $b < -r \le \mathbf{a}^\top \mathbf{x}$. This causes a contradiction since $\mathbf{x} \in \mathcal{D}$ is a feasible point satisfying the linear inequality $\mathbf{a}^\top \mathbf{x} \le \mathbf{b}$.

Dividing both sides by $r$ gives

$$\frac{p}{r} \cos\theta + \frac{q}{r} \sin\theta = \frac{b}{r}.$$

There exists a unique angle $\tau \in [-\pi, \pi]$ (ignoring the repetition at the boundary) such that $\cos\tau = \frac{p}{r}$ and $\sin\tau = \frac{q}{r}$. In practice, $\tau$ is given by `arctan2(q, p)`, a function implemented in many libraries. Applying the angle sum formula gives

$$\cos(\theta - \tau) = \frac{b}{r}.$$

It is clear that the ratio $b/r$ determines the number of roots. When $-1 < b/r < 1$, the two distinct roots are given by

$$\theta = \tau \pm \arccos\left(\frac{b}{r}\right), \tag{4}$$

A multiple of $2\pi$ has to be added to the roots, if necessary, to make sure the angles fall in into $[0, 2\pi]$.

Note that (4) is not the only form of the roots. For instance, Gessner et al. (2020) used the roots of the form

$$\theta = \pm \arccos\left(\frac{b}{r}\right) + 2\arctan\left(\frac{q}{r+p}\right).$$

Another root formula, used by Benavoli et al. (2021), is of the form

$$\tan\frac{1}{2}\theta = \frac{q \pm \sqrt{r^2 - b^2}}{b + p}.$$

Proving these root formulas is left as an exercise for the readers. Despite their equivalence, we recommend using our root formula (4). This is an unbiased opinion, since we arrive at this conclusion after trying all formulas. The other two root formulas need to check additional edge cases when $r + p \approx 0$ and $b + p \approx 0$, which do happen annoyingly in certain extreme situation in practice.

## C  Proofs

**Proposition 1.** *Let $\alpha_i < \beta_i$ for all $i \in [m]$ and let $\{\alpha_{i_k}\}_{k=1}^m$ be sorted in ascending order as in (3). Then, the active intervals $I_{\text{act}} = \cap_{i=1}^m \big([0, \alpha_i] \cup [\beta_i, 2\pi]\big)$ have an equivalent representation*

$$I_{\text{act}} = [0, \alpha_{i_1}] \cup \left(\bigcup_{k=2}^m [\gamma_{k-1}, \alpha_{i_k}]\right) \cup [\gamma_m, 2\pi],$$

*where $\gamma_k = \max\{\beta_{i_1}, \beta_{i_2}, \cdots, \beta_{i_k}\}$ is the cumulative max until $\beta_{i_k}$. We interpret the interval $[\gamma_{k-1}, \alpha_{i_k}]$ as an empty set if $\gamma_{k-1} > \alpha_{i_k}$.*

*Proof.* The sorted angles $\{\alpha_{i_k}\}_{k=1}^m$ divide $[0, 2\pi]$ into $m+1$ disjoint segments:
$$[0, 2\pi] = [0, \alpha_{i_1}] \cup (\alpha_{i_1}, \alpha_{i_2}] \cup \cdots \cup (\alpha_{i_{m-1}}, \alpha_{i_m}] \cup (\alpha_{i_m}, 2\pi].$$
The active intervals $I_{\text{act}}$ are constructed by computing the intersection of $I_{\text{act}}$ with each segment. That is, we use the trivial identity
$$I_{\text{act}} = I_{\text{act}} \cap [0, 2\pi] = \underbrace{\left(I_{\text{act}} \cap [0, \alpha_{i_1}]\right)}_{\text{part one}} \cup \underbrace{\left(\bigcup_{k=1}^{m-1} \left(I_{\text{act}} \cap (\alpha_{i_k}, \alpha_{i_{k+1}}]\right)\right)}_{\text{part two}} \cup \underbrace{\left(I_{\text{act}} \cap [\alpha_{i_m}, 2\pi]\right)}_{\text{part three}}$$
and compute each part analytically.

**Part One.** The intersection with the first segment is easy to compute:
$$I_{\text{act}} \cap [0, \alpha_{i_1}] = \left(\bigcap_{i=1}^m I_i\right) \cap [0, \alpha_{i_1}] = \bigcap_{i=1}^m (I_i \cap [0, \alpha_{i_1}]) = [0, \alpha_{i_1}],$$
where the first equality uses the definition of the active intervals $I_{\text{act}}$; the second equality swaps the order of intersections; the third equality uses this observation: $[0, a_{i_1}]$ is a subset of all $I_i$ for $i \in [m]$ since $\alpha_{i_1}$ is the smallest angle among all $\alpha_i$'s and $\beta_i$'s.

**Part Three.** Similarly, the intersection with the last segment is also easy to compute:
$$I_{\text{act}} \cap (a_{i_m}, 2\pi] = \bigcap_{i=1}^m \left(I_i \cap (\alpha_{i_m}, 2\pi]\right) = \bigcap_{i=1}^m \left([\beta_i, 2\pi] \cap (\alpha_{i_m}, 2\pi]\right) = \left[\max_{1 \leq i \leq m} \beta_i, 2\pi\right],$$
where the first equality plugs in the definition $I_{\text{act}} = \cap_{i=1}^m I_i$; the second equality is because $\alpha_{i_m}$ is the largest angle among all $\alpha_i$'s and therefore we can ignore $[0, \alpha_i]$; the last equality is due to
$$\alpha_{i_m} < \beta_{i_m} \leq \max_{1 \leq i \leq m} \beta_i$$
and thus the chunk $[\alpha_{i_m}, \max_{1 \leq i \leq m} \beta_i)$ is removed from $[a_{i_m}, 2\pi]$.

**Part Two.** Now we deal with the remaining segments $(a_{i_{k-1}}, a_{i_k}]$ for $k = 2, 3, \cdots, m$. We assume $a_{i_{k-1}}$ is strictly smaller than $a_{i_k}$ for now and defer the case $a_{i_{k-1}} = a_{i_k}$ to the end. For a fixed $k$, we must compute
$$I_{\text{act}} \cap (a_{i_{k-1}}, a_{i_k}] = \left(\bigcap_{i=1}^m I_i\right) \cap (a_{i_{k-1}}, a_{i_k}] = \bigcap_{i=1}^m \left(I_i \cap (a_{i_{k-1}}, a_{i_k}]\right).$$
Now we split the intersection index $i$ into two cases $\{i \in [m] : \alpha_i \geq \alpha_{i_k}\}$ and $\{i \in [m] : \alpha_i \leq \alpha_{i_{k-1}}\}$. For the first case, notice that
$$\bigcap_{\{i \in [m] : \alpha_i \geq \alpha_{i_k}\}} \left(I_i \cap (a_{i_{k-1}}, a_{i_k}]\right) = (\alpha_{i_{k-1}}, \alpha_{i_k}],$$
because the choice of the index $i$ implies $(\alpha_{i_{k-1}}, \alpha_{i_k}] \subseteq [0, \alpha_i] \subseteq I_i$. For the second case, we have
$$\bigcap_{\{i \in [m] : \alpha_i \leq \alpha_{i_{k-1}}\}} \left(I_i \cap (a_{i_{k-1}}, a_{i_k}]\right) = \bigcap_{\{i \in [m] : \alpha_i \leq \alpha_{i_{k-1}}\}} \left(([0, \alpha_i] \cup [\beta_i, 2\pi]) \cap (\alpha_{i_{k-1}}, \alpha_{i_k}]\right)$$
$$= \bigcap_{\{i \in [m] : \alpha_i \leq \alpha_{i_{k-1}}\}} \left([\beta_i, 2\pi] \cap (\alpha_{i_{k-1}}, \alpha_{i_k}]\right)$$
$$= \left[\max\{\beta_{i_1}, \beta_{i_2}, \cdots, \beta_{i_{k-1}}\}, 2\pi\right] \cap (\alpha_{i_{k-1}}, \alpha_{i_k}],$$
where the first equality plugs in the definition of $I_i$; the second equality is because the index $i$ is specifically chosen such that $\alpha_i \leq \alpha_{i_{k-1}}$; the last equality is because $\alpha_i$'s are sorted: the indices such that $\alpha_i \leq \alpha_{i_{k-1}}$ are precisely $i_1, i_2, \cdots, i_{k-1}$.

Define $\gamma_k = \max\{\beta_{i_1}, \beta_{i_2}, \cdots, \beta_{i_{k-1}}\}$. Combining the two cases, we obtain
$$I_{\text{act}} \cap (a_{i_{k-1}}, a_{i_k}] = \begin{cases} [\gamma_k, \alpha_{i_k}] & \text{if } \gamma_k \leq \alpha_{i_k} \\ \emptyset & \text{otherwise} \end{cases}$$
for $k = 2, 3, \cdots, m$. Finally, we come back to the edge case $a_{i_{k-1}} = a_{i_k}$, which implies $(a_{i_{k-1}}, a_{i_k}] = \emptyset$ by convention. Thus, the intersection $I_{\text{act}} \cap (a_{i_{k-1}}, a_{i_k}]$ is automatically empty. One can verify that the above expression outputs an empty set as well, since $\gamma_k \geq \beta_{i_{k-1}} > \alpha_{i_{k-1}} = \alpha_{i_k}$. $\quad\square$

---

**Algorithm 3:** Constructing the Active Intervals by Brute Force

**Input:** $I_i = [0, \alpha_i] \cup [\beta_i, 2\pi]$ for $i = 1, 2, \cdots, m$
**Output:** $I_{\mathrm{act}} = \cap_{i=1}^m I_i$

**1** $I_{\mathrm{act}}^{(0)} = [0, 2\pi]$
**2 for** $i = 1, 2, \cdots, m$ **do**
**3** $\quad \Big| \quad I_{\mathrm{act}}^{(i)} = I_{\mathrm{act}}^{(i-1)} \cap \big( [0, \alpha_i] \cup [\beta_i, 2\pi] \big)$
**4 end**
**5 return** $I_{\mathrm{act}}^{(i)}$

---

## D    Brute Force Intersection Time Complexity

This section shows that the time complexity of Algorithm 3 is at least $\Omega(m^2)$. To do so, we construct a worst-case input on which Algorithm 3 takes at least $\Omega(m^2)$ operations.

Without loss of generality, we will work with intervals of the form $I_i = [0, \alpha_i] \cup [\beta_i, 1]$. Consider the following intersection intervals

$$I_i = \left[ 0, \left( \frac{1}{3} \right)^i \right] \cup \left[ 2 \left( \frac{1}{3} \right)^i, 1 \right], \quad i = 1, 2, \cdots, m.$$

By induction, it is easy to show that the intersection of the first $k$ intervals is

$$\bigcap_{i=1}^k I_i = \left[ 0, \left( \frac{1}{3} \right)^k \right] \cup \left( \bigcup_{i=1}^k \left[ 2 \left( \frac{1}{3} \right)^i, \left( \frac{1}{3} \right)^{i-1} \right] \right).$$

In particular, $\cap_{i=1}^k I_i$ is an union of $k + 1$ intervals. Thus, the $k$-th inner loop of Algorithm 3 takes $\Omega(k)$ operations. As a result, the algorithm takes $\Omega(m^2)$ operations in total. It is also not hard to verify that presorting $\{\alpha_i\}_{i=1}^m$, or $\{\beta_i\}_{i=1}^m$, does not reduce the time complexity. The worse-case inputs for these variants can be constructed similarly.

## E    Additional Experimental Details

The follow table presents the statistics of the MCMC runs in §4.

| truncation | $[-1, 3]$ | $[15, 16]$ |
|---|---|---|
| true $\mu$ | 0.2828 | 15.0661 |
| estimates $\hat{\mu}$ | 0.2820 | 15.0661 |
| true $\sigma^2$ | 0.6161 | 0.0043 |
| estimates $\hat{\sigma}^2$ | 0.6148 | 0.0043 |

