# OpenReview forum: "A Fast, Robust Elliptical Slice Sampling Method for Truncated Multivariate Normal Distributions"
_NeurIPS.cc/2024/Workshop/BDU — NeurIPS BDU Workshop 2024 Poster_

### Official Review · Reviewer_LJqW · 2024-09-19
**The review identifies key areas for improvement in the paper, including the need for stronger theoretical justification, more comprehensive experimental validation with real-world data, and broader comparisons with alternative methods. The paper could benefit from a deeper analysis of accuracy, convergence, and numerical stability, as well as a clearer explanation of the algorithm’s pseudocode and potential GPU performance limitations.**

**Rating:** 5
**Confidence:** 5

**Review:**

1. While the paper introduces a faster \(O(m \log m)\) algorithm for the ellipse-polytope intersection, the theoretical justification for this improvement could be strengthened. There is a lack of detailed analysis explaining why this method significantly outperforms existing \(O(m^2)\) algorithms beyond empirical results.

2. The experiments focus mainly on synthetic datasets with large dimensions, which does validate the speed-up, but they are not very diverse. Testing the algorithm on real-world datasets or applications would provide a better understanding of its practical relevance. The lack of results from domains such as Bayesian optimization or Gaussian processes, which are cited as use cases, leaves a gap in practical validation.

3. The comparison is limited to BoTorch's elliptical slice sampling implementation. While this is useful, it would be more informative to compare the new algorithm against other alternative methods, such as rejection sampling or Hamiltonian Monte Carlo (HMC). This would provide a broader context for understanding the algorithm’s efficiency, especially in scenarios where other methods might also be applied.

4. The paper focuses primarily on speed improvements but provides little discussion on the accuracy and convergence properties of the algorithm. How does the method ensure that the samples drawn are representative of the truncated multivariate normal distribution?

5. The \(O(m \log m)\) time complexity is theoretically advantageous, but the practical implementation might still face bottlenecks depending on the nature of the constraints. The authors could clarify under which conditions the method performs optimally and when it might face challenges.

6. The authors briefly mention other slice sampling methods in the introduction but do not explore how their method compares in terms of complexity, ease of implementation, and adaptability.

7. While the paper shows promising results using GPU parallelism, there is no discussion on the limitations of GPU scalability. How well does the method scale when moving from consumer-grade GPUs, like the RTX 3090, to larger setups?

8. The paper touches on numerical stability in the introduction but does not delve into how the new algorithm ensures stability in practice, especially in high-dimensional cases. Numerical instabilities, particularly when handling many constraints, can cause issues in Monte Carlo methods.

9. Although the pseudocode in Algorithm 2 is concise, the explanation accompanying it lacks detail in certain areas. For example, the transition between sorting the angles and computing the cumulative max is under-explained.

---

### Official Review · Reviewer_8n3A · 2024-09-25
**The paper proposed a O(n logn) algorithm for computing the analytical active intervals in linear elliptical slice sampling.**

**Rating:** 7
**Confidence:** 3

**Review:**

**Pros:**

1. The manuscript is clearly written, easy to understand, and the proposed method is straightforward to implement.
2. The proofs are well-structured and of high quality.
3. The method effectively improves the complexity of high-dimensional slice sampling, which is a significant contribution for MCMC method.


**Questions and Suggestions:**

1. On line 15, the authors said the polytope is not necessarily bounded, it's a confusing claim. I think the definition of  "polytope" is the bounded (and convex) polyhedron. If the authors only refer to the solution of a set of a finite number of linear constraints, "polyhedron" might be a more acccurate term.

2. Is there a way to improve the algorithm when $m = d$, where $m$ is the number of linear constraints and $d$ is the dimension? It seems that inactive intersections mainly or only occur when $m > d$, i.e., the matrix $A$ is not full-rank.

3. In Section 4.1, the authors evaluate the performance of slice sampling in a one-dimensional case. Given that the focus of the manuscript is on multi-dimensional sampling, it would be more informative to assess the method's performance in two-dimensional cases.

4. In Section 4.2, the polytope with non-empty interior is constructed by generating a random matrix, adding a nonzero uniformly distributed random vector, and setting $m = d$. Does this setup tend to create a polytope with certain properties? For instance, if matrix $A$ is filled with i.i.d. standard normal random variables, it is almost always full rank. Could this influence the structure of the active and inactive intervals? If so, it would be helpful to mention this or include a broader range of polytopes in the experiment. As I mentioned above, checking inactive intersections is the time-consuming part of computing the analytical active intervals, and this seems to occur primarily, or only, when $m > d$ (i.e., when matrix $A$ is not full rank). Therefore, designing an experiment to show how the number of inactive intersections affects the efficiency of standard elliptical slice sampling, and how the proposed method speeds up this process, would better highlight the advantages of the algorithm.

5. I think it would help demonstrate the algorithm’s benefits in real-world scenarios by including some practical applications, such as Bayesian optimization. For instance, in what contexts would sampling from a 1000-dimensional truncated multivariate normal distribution be necessary? Mostly we don't have to sample from a real high dimensional space, therefore giving some real cases would be more convincinng.

---

### Decision · Program_Chairs · 2024-10-09

Accept (Poster)